

# Prediction of sea ice area based on the CEEMDAN-SO-BiLSTM model

Qiao Guo, Haoyu Zhang, Yuhao Zhang and Xuchu Jiang

Zhongnan University of Economics and Law, Wuhan, Hubei, China

## ABSTRACT

This article proposes a combined prediction model based on a bidirectional long short-term memory (BiLSTM) neural network optimized by the snake optimizer (SO) under complete ensemble empirical mode decomposition with adaptive noise. First, complete ensemble empirical mode decomposition with adaptive noise (CEEMDAN) was used to decompose the sea ice area time series data into a series of eigenmodes and perform noise reduction to enhance the stationarity and smoothness of the time series. Second, this article used a bidirectional long short-term memory neural network optimized by the snake optimizer to fully exploit the characteristics of each eigenmode of the time series to achieve the prediction of each. Finally, the predicted values of each mode are superimposed and reconstructed as the final prediction values. Our model achieves a good score of RMSE: 1.047, MAE: 0.815, and SMAPE: 3.938 on the test set.

## INTRODUCTION

Sea ice extent is a key observational indicator of climate change and diversity (*Serreze, Holland & Stroeve, 2007*) Over the past half-century, satellite observations have revealed a gradual increase in Arctic temperatures, a gradual decrease in sea ice cover, and the emergence of Arctic amplification (*Holland et al., 2019*). At different time scales, sea ice cover anomalies can have extreme effects on atmospheric circulation and precipitation patterns, which in turn can further affect the climate at mid- and high latitudes (*Screen, 2013*), such as the 2021 cold snap in Texas and Oklahoma. Based on current trends, Arctic sea ice could disappear completely by 2050 (*Notz & Stroeve, 2018*). In addition, accurate daily, quarterly and annual monitoring and prediction of changes in sea ice extent have important implications for human exploitation of maritime resources, navigation activities in sea ice regions, and global climate analysis and prediction (*Smith & Stephenson, 2013*; *Choi, De Silva & Yamaguchi, 2019*; *Cavalieri et al., 1999*). Therefore, accurate prediction of sea ice movement is essential for human activity and climate modeling (*Stroeve et al., 2012*).

At present, research on modeling the characteristics of sea ice mainly involves statistical models and numerical models. Statistical models are constructed based on historical observations and relationships between atmospheric conditions (*e.g.*, temperature, sea level pressure, and clouds), ocean conditions (*e.g.*, sea surface temperature), and sea ice variables

Corresponding author
Xuchu Jiang, xuchujiang@zuel.edu.cn

(*e.g.*, concentration, extent, ice type, and thickness). For example, *Turner et al. (2013)* used statistical regression to analyze the relationship between the Amundsen Sea low pressure and Antarctic Sea ice cover, indicating that the deepening of the Amundsen Sea low pressure is associated with West Antarctic warming and the expansion of the Ross Sea ice cover. However, *Wang, Chen & Kumar (2013)* believe that statistical methods do not consider the interaction between sea ice and the atmosphere, and there are certain limitations. Numerical models are primarily physically driven models based on the physical equations of control system dynamics and thermodynamics, such as *Gent et al. (2011)*, which describe all developments in the Community Climate System Model (CCSM) and document fully coupled preindustrial control operations compared to previous versions of CCSM3. *Guemas et al. (2016)* argue that numerical models are generally superior to statistical models in short-term forecasting. However, while inputs such as atmosphere, oceanic and ice parameters can be obtained from remotely sensed data, they must be calibrated and validated through spatially and temporally well-distributed *in situ* observations, which are difficult and costly to obtain and therefore often inefficient.

Machine learning and deep learning techniques have developed rapidly in recent years and have shown significant advantages in sea ice cover prediction. *Barnhart et al. (2016)* used support vector machine (SVM) models to analyze the relationship between sea ice and climate variables, successfully predicting Arctic open water expansion and Arctic sea ice changes in the coming decades. Deep learning model prediction also solves the limitation of numerical models in multiparameter accurate acquisition to some extent (*Rasp, Pritchard & Gentine, 2018*), but there are still some challenges in capturing temporal correlations in the time series prediction of nonlinear sea ice area data (*Ren, Li & Zhang, 2022*). The LSTM model has attracted great attention due to the rapid development of artificial intelligence and its ability to automatically extract feature modeling (*Hochreiter & Schmidhuber, 1997*), and the research of *Siami-Namini, Tavakoli & Namin (2019)* also proved that the predictive performance of bidirectional LSTM is due to LSTM. With the study of time series frequency domain analysis methods, the EMD method (*Huang et al., 1998*) was developed, which decomposes noisy data according to its own time scale characteristics and does not need to set any basis function in advance, which has obvious advantages in processing nonstationary and nonlinear data. *Torres et al. (2011)* proposed the adaptive noise complete set empirical modal decomposition (CEEMDAN) algorithm, which overcame the defects of EMD and EEMD decomposition loss of completeness and modal aliasing by adaptive addition of white noise. *Hu et al. (2022)* integrated CEEMDAN with LSTM and temporal convolutional networks (TCN) to enable ultra-short-term wind power forecasting and real-time prediction of wind energy. Similarly, *Gao & Zhang (2023)* leveraged a combined approach of variational mode decomposition (VMD) and LSTM for decomposed prediction. Their research focused on the impact of investor sentiment on price volatility in China's capital market. The results from both studies underscore the efficacy of such combined methodologies in their respective fields.

Therefore, this article explored an optimal data-driven time series model combining the empirical mode decomposition (EMD) method and optimized deep learning neural networks to capture the nonlinear and nonstationary characteristics of sea ice area time

series. This combination allows us to better understand the temporal correlations present in the sea ice area series and overcome the limitations of current time series models. This article compared the performance of our proposed model with both a benchmark model and similar approaches and analyze their differences and advantages. Our analysis demonstrates the superiority and effectiveness of our target model.

# THEORETICAL MODEL CONSTRUCTION

## Complete ensemble empirical mode decomposition with adaptive noise (CEEMDAN)

Due to the "mode mixing" caused by EMD and the noise residual caused by EEMD, this article introduces complete ensemble empirical mode decomposition with adaptive noise (CEEMDAN), which overcomes the defects of EEMD decomposition in terms of loss of completeness and mode mixing by adaptively adding white noise. In the algorithm, $E_i(\cdot)$ is defined as the $i$-th mode generated by EMD decomposition, $\overline{Ci(t)}$ represents the ith mode generated by CEEMDAN decomposition, $\varepsilon$ is the standard deviation of the noise, $v^j$ follows $N(0,1)$ and j =1,2, ....,N denotes the number of times white noise is added, while $r$ represents the residue. The specific steps of the CEEMDAN algorithm are as follows:

(A) Add Gaussian white noise to the original signal $y(t)$ to obtain a new signal $y(t)+(-1)^q \varepsilon v^j$, where $q = 1,2$. EMD is performed on the signal to obtain the first-stage intrinsic mode component $C_1$

$$E\left(y(t)+(-1)^q \varepsilon v^j(t)\right) = C_1^j(t)+r^j \qquad (1)$$

(B) Taking the overall average of the $N$ generated mode components produces the first intrinsic mode function of the CEEMDAN decomposition.

$$\overline{C_1(t)} = \frac{1}{N}\sum_{j=1}^{N} C_1^j(t) \qquad (2)$$

(C) Calculate the residue after removing the first modal component

$$r_1(t) = y(t) - \overline{C_1(t)} \qquad (3)$$

(D) Add paired positive and negative Gaussian white noise to $r_1(t)$ to obtain a new signal, and perform EMD on the new signal to obtain the first-order modal component $D_1$. Then, the second intrinsic mode component of the CEEMDAN decomposition can be obtained.

$$\overline{C_2(t)} = \frac{1}{N}\sum_{j=1}^{N} D_1^j(t) \qquad (4)$$

(E) Calculate the residue after removing the second modal component

$$r_2(t) = r_1(t) - \overline{C_2(t)} \qquad (5)$$

(F) The above steps are repeated until the obtained residual signal is a monotonic function and cannot be further decomposed, and the algorithm ends. The number of intrinsic mode functions obtained at this time is denoted as $K$, and the original signal y(t) can be decomposed as:

$$y(t) = \sum_{k=1}^{K} \overline{C_k(t)} + r_k(t) \qquad (6)$$
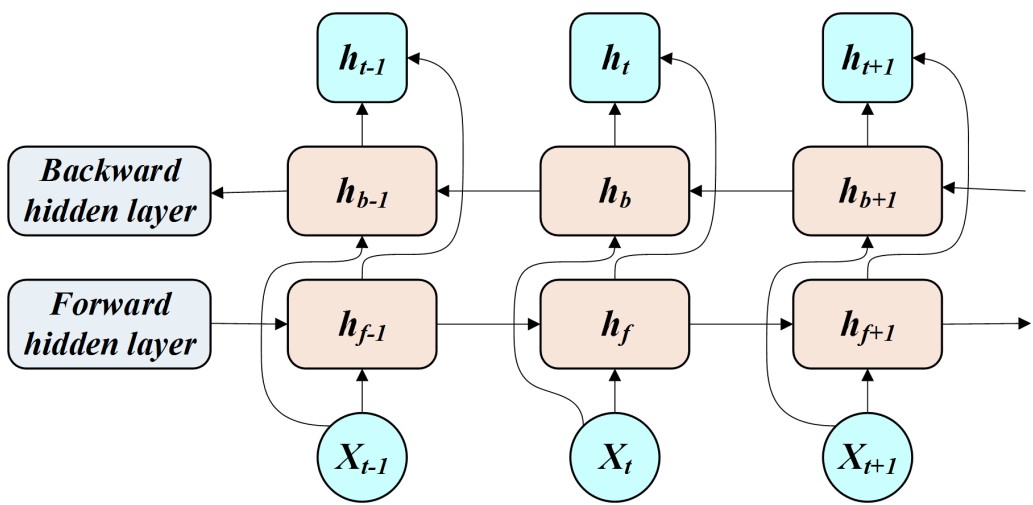

**Figure 1** Working process of BILSTM.

## Bidirectional long short-term memory neural network (BiLSTM) and optimization

### BiLSTM

The BiLSTM allows for the transmission and feedback of past and future states of the hidden layers through a bidirectional network (Fig. 1).

The BiLSTM is interpreted through Eq. (7).

$$\begin{cases} h_f = \text{LSTM}(x_i, h_{f-1}) \\ h_b = \text{LSTM}(x_t, h_{b-1}) \\ h_t = w_t h_f + v_t h_b + b_t \end{cases} \tag{7}$$

where $x_i$ is the input, $h_f$ is the forward-pass implicit layer state, $h_b$ is the reverse-pass implicit layer state, $h_t$ is the implicit layer state, $w_t$ is the forward-pass implicit layer output weight, $v_t$ is the reverse-pass implicit layer output weight, and $b_t$ is the error value.

### Adaptive moment estimation (Adam)

Adam's algorithm improves model accuracy and network training speed by calculating the first-order moments and second-order moment estimates that can be adapted to the corresponding learning rate by computing the gradient of the objective function. Each iteration of Adam's update of the BiLSTM parameter $\theta_t$ is

$$\theta_t = \theta_{t-1} - \alpha \frac{\hat{m}_t}{\sqrt{\hat{n}_t} + \varepsilon} \tag{8}$$

where $\hat{m}_t$ and $\hat{n}_t$ are the corrected first- and second-order moment estimates, respectively, and $\varepsilon$ is a constant $10^{-8}$.

### BiLSTM for snake optimizer optimization (SO-BiLSTM)

Based on the special mating behavior of snakes, *Hashim & Hussien (2022)* proposed the snake optimizer (SO). The algorithm is divided into two stages: global exploration when

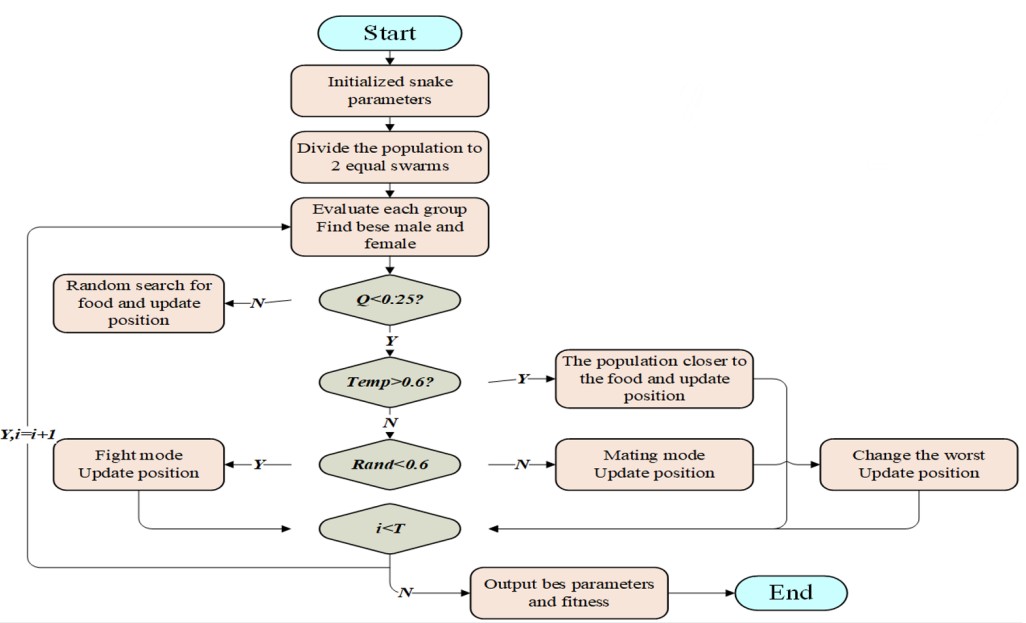

**Figure 2** Working process of SO.

there is no food and local exploitation when there is food. When food is scarce ($Q<0.25$), the snakes search for food and update the location by choosing any random location to achieve global optimization. When there is sufficient food ($Q \geq 0.25$), snakes move toward food and update their position if $Temp>0.6$; if the temperature $Temp \leq 0.6$, the snakes enter combat mode at a random number $Rand<0.6$ taking the value of [0,1] and enter mating mode at $Rand \geq 0.6$ and replace the individual snake with the worst fitness value by laying and hatching eggs.

The prediction accuracy of the BiLSTM neural network is affected by the number of neurons in the hidden layer, the learning rate and the L2 regularization coefficient. Considering the randomness of artificially set parameters, the root mean square error RMSE is selected as the fitness function in this article, and the parameters are optimized using the SO algorithm (Fig. 2).

## CEEMDAN-SO-BiLSTM

Through the above analysis, this article introduces the CEEMDAN module for noise reduction of the original data based on the advantages of the optimization algorithm and deep learning and proposes a combined CEEMDAN-SO-BiLSTM prediction model (Fig. 3).

As seen from the above figure, the prediction model in this article is divided into three parts. The first part is CEEMDAN decomposition, which decomposes the time series into $K$ modes for noise reduction; the second part is the SO-BiLSTM neural network model to train the prediction of $K$ modes; and the third part superimposes and reconstructs the prediction results of the second part to obtain the prediction results of the original data.

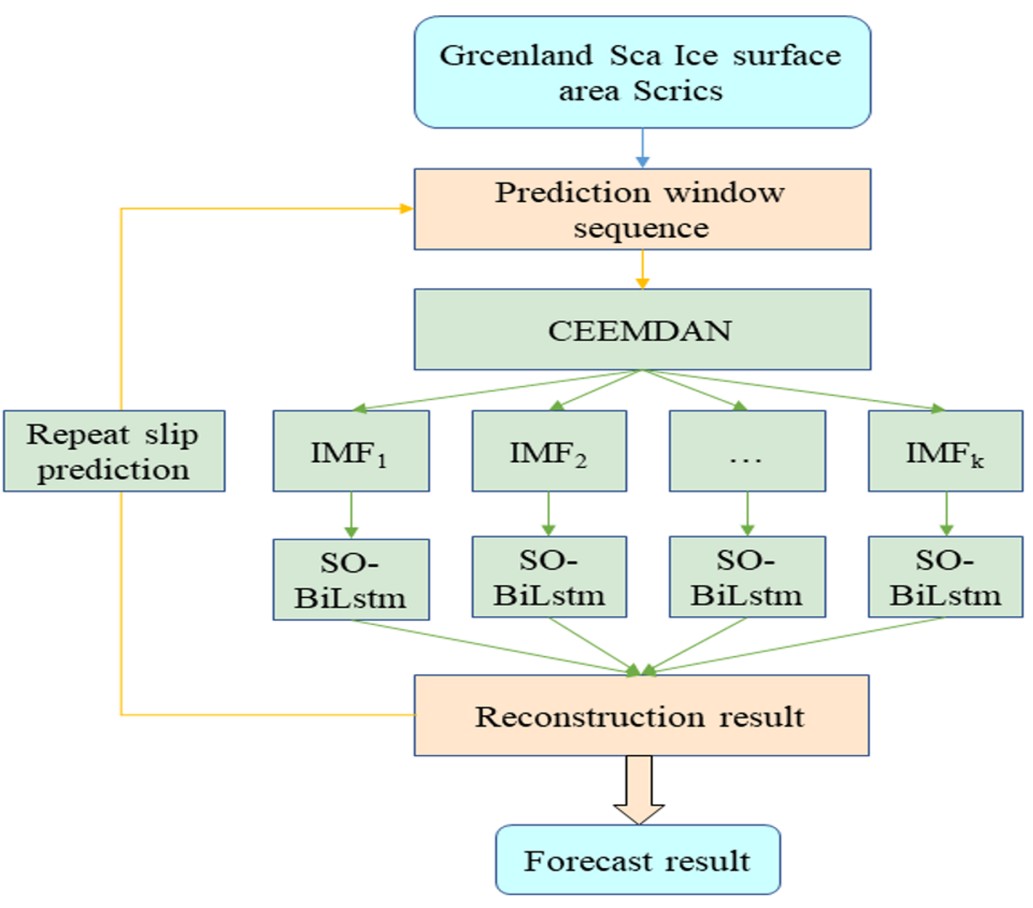

**Figure 3** **Working process of CEEMDAN-SO-BILSTM.**

## EXPERIMENTAL ANALYSIS

### Data sources

The data set used in this article was obtained from the sea ice data set on the official website of the National Environmental Information Center (https://www.ncei.noaa.gov/) and the Grambling Sea Ice area time series data was used for the study in the model analysis. The time range is from day 272 of 2017 to day 271 of 2022, with a total of 1,824 data points.

### Model evaluation criteria

To evaluate the prediction performance of the model, this article selects four indicators: mean absolute error (MAE), root mean square error (RMSE), determinability coefficient $R^2$, and symmetric mean absolute percentage error (SMAPE). Equations for calculating these indicators are provided in Eqs. (9)–(12).

These indicators quantify the accuracy of the model's predictions. MAE and RMSE represent the average and root of the squared errors between the predicted and actual values, respectively, while $R^2$ measures the correlation between the predicted and actual values. SMAPE calculates the symmetric percentage difference between the predicted and

actual values.

$$\text{MAE} = \frac{1}{n}\sum_{i=1}^{n}|y_i - \hat{y}_i| \tag{9}$$

$$\text{RMSE} = \sqrt{\frac{1}{n}\sum_{i=1}^{n}(y_i - \hat{y}_i)^2} \tag{10}$$

$$\text{R}^2 = \frac{\sum_{i=1}^{n}(y_i - \hat{y}_i)^2}{\sum_{i=1}^{n}(y_i - \bar{y})^2} \tag{11}$$

$$\text{SMAPE} = \frac{100\%}{n}\sum_{i=1}^{n}\frac{|\hat{y}_i - y_i|}{(|\hat{y}_i| + |y_i|)/2} \tag{12}$$

where $y_i$ is the $i$-th observation, $\hat{y}_i$ is the $i$-th prediction, $\bar{y}$ is the mean, and $n$ is the number of samples.

## Experimental analysis

According to the characteristics of the data sets, this article first partitions the original data into a training set comprising 70% of the data and a testing set consisting of the remaining 30%. Then, using the rolling window method with CEEMDAN decomposition, it predicts the values within each roll-out window. The noise ratio is set at 0.2, with 500 iterations of adding noise to the signal and a maximum span of 2000. For example, it shows the different modalities obtained by decomposing the first prediction window (Fig. 4), including 10 modes representing various frequency dimensions of the ice concentration time series. The high correlation coefficients of IMF8 and IMF9 with the original sequence, reaching 0.91 and 0.77, respectively, indicate that these two modes contribute significantly to the periodic trend of the original signal. IMF1 to IMF3 are considered as noise signals, while IMF10 represents a short-term trend component (Fig. 5). As seen from the figure, compared to the original sequence, the decomposed modalities are more stable and smooth and exhibit clear information features, providing a solid foundation for predictions.

The specific process of making predictions using the rolling window method is as follows: assuming that the model training and test time series data are D, with length T + k, where the first T data points are used for model training and the last k data points are used for model testing. It uses the previous 30 data points to predict the next data point in the testing set. To obtain each predicted value from the testing set, follow these steps:

1. CEEMDAN decomposition is performed on the first T data points to extract multiple features and train separate models for each feature.
2. Following completion of model training, utilize data from D[T-29:T] to obtain predicted outputs from each feature model and sum them to acquire the prediction value for D[T+1].

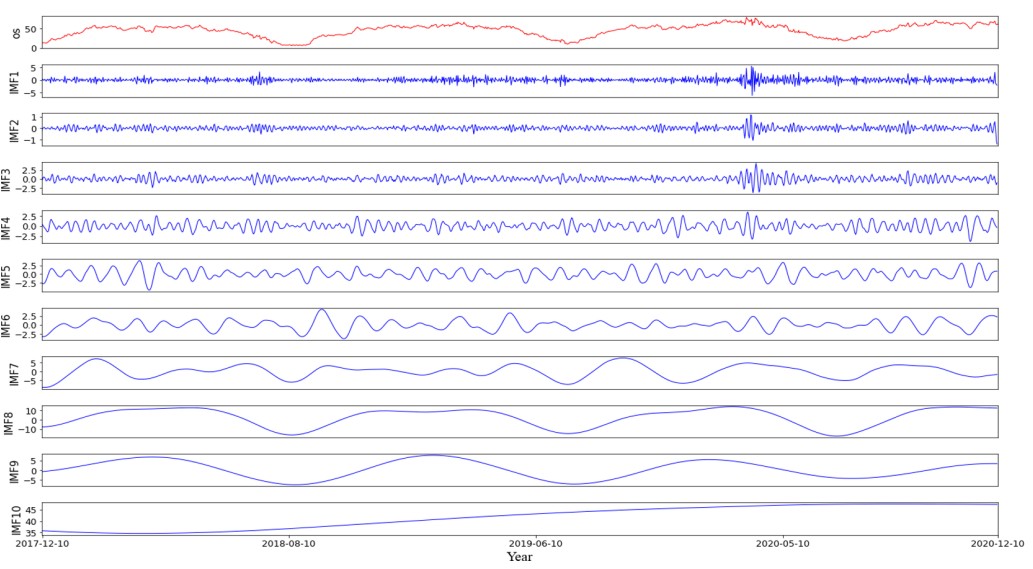

**Figure 4  Results of empirical modal decomposition of the original sequence adaptive noise complete set.**

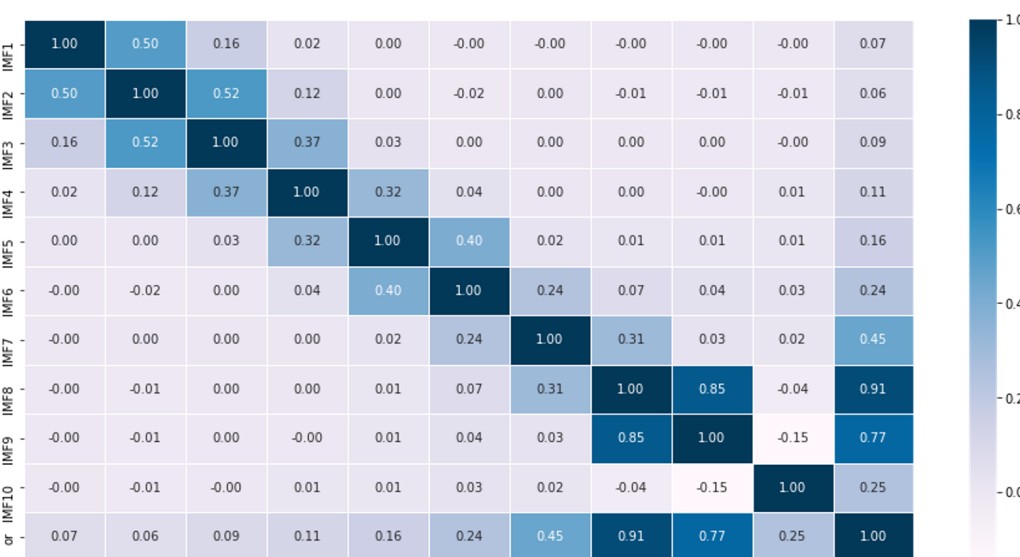

**Figure 5  Adaptation curve of SO.**

3. Execute CEEMDAN1 decomposition on data D[2:T+1] to extract multiple features once again and train individual models for every feature. Utilizing 30 data points from D[T-28:T+1], it receives the predicted data from each feature model and aggregate them to attain the forecast for D[T+2].

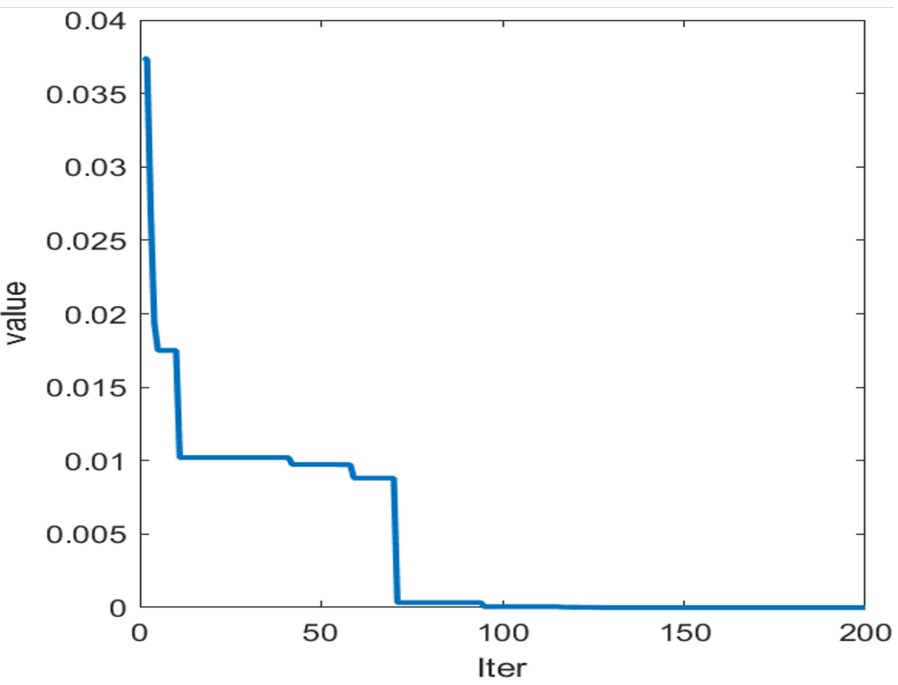

**Figure 6** **Prediction results of each mode.**

Repeat actions 1 through 3, sliding the window of 30 data points and performing decomposition and model training on the enclosed data before making predictions, ultimately arriving at projections for all future time intervals D[T+1:T+k].

After applying SO-BiLSTM training for predictive outcomes from the preceding splitting apart, SO adjusts BiLSTM's primary acquisition pace, opaque node contingent, and L2 regulation factor. Following numerous trials, the versatile extent characteristic ultimately slopes toward equilibrium (Fig. 6), and the versions for every mode within the initial estimation aperture are improved as per Table 1.

For each modality, the subsequent predictive value is recreated so that the concluding forecast can be generated for the initial information. By persistently pushing the prognostication casement ahead and producing supplementary projections, up to thirty percent of all estimations have been accomplished (Fig. 7). Evaluation parameters for the objective model are determined using Table 2.

## Model comparison

To verify the superiority of the model proposed in this article, models such as BiGRU and BiLSTM were tested and compared with the model in this article, and the test results of each model are shown in Table 3.

Based on our findings (Fig. 8), it appears that the performance of the comparison model deteriorates during specific periods, particularly around May 2021 and January 2022, characterized by substantial fluctuations and increased prediction bias. Notably, the

**Table 1  Prediction hyperparameters of each mode.**

| Modal | Number of hidden layer nodes | Initial learning rate | L2 regularization factor |
|---|---|---|---|
| IMF1 | 108 | 0.013 | $10^{-9}$ |
| IMF2 | 96 | 0.017 | $10^{-10}$ |
| IMF3 | 97 | 0.018 | $10^{-8}$ |
| IMF4 | 100 | 0.001 | $10^{-8}$ |
| IMF5 | 200 | 0.012 | $10^{-10}$ |
| IMF6 | 200 | 0.001 | $10^{-10}$ |
| IMF7 | 100 | $10^{-4}$ | $10^{-10}$ |
| IMF8 | 100 | $10^{-4}$ | $10^{-10}$ |
| IMF9 | 100 | $10^{-4}$ | $10^{-10}$ |
| IMF10 | 100 | $10^{-4}$ | $10^{-10}$ |

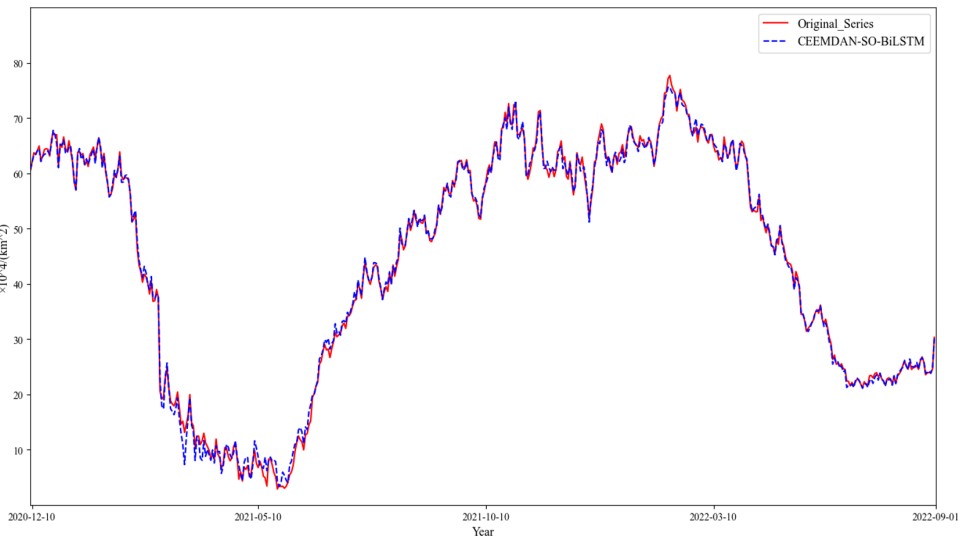

**Figure 7  Reconfiguration results.**

**Table 2  CEEMDAN-SO-BiLSTM prediction effects.**

| Evaluation indicators | Indicator value |
|---|---|
| MAE | 0.815 |
| RMSE | 1.047 |
| $R^2$ | 0.998 |
| SMAPE | 3.938% |

**Table 3  Comparison of the predicted effects of the original data.**

| Models | MAE | RMSE | R$^2$ | SMAPE |
|---|---|---|---|---|
| ARIMA | 5.029 | 6.229 | 0.913 | 18.456% |
| SVR | 4.467 | 5.649 | 0.929 | 17.154% |
| BiLSTM | 3.323 | 4.300 | 0.961 | 13.231% |
| BiGRU | 3.726 | 4.739 | 0.950 | 14.688% |
| CEEMDAN-BiGRU | 2.930 | 3.574 | 0.971 | 11.986% |
| CEEMDAN-BiLSTM | 2.583 | 3.248 | 0.976 | 11.848% |
| VMD-BiGRU | 2.610 | 3.586 | 0.971 | 13.050% |
| VMD-BiLSTM | 2.605 | 3.561 | 0.972 | 12.944% |
| VMD-SO-BiLSTM | 2.021 | 4.182 | 0.981 | 10.424% |
| CEEMDAN-SO-BiGRU | 1.917 | 2.783 | 0.983 | 10.137% |
| CEEMDAN-SO-LSTM | 1.828 | 2.416 | 0.987 | 9.060% |

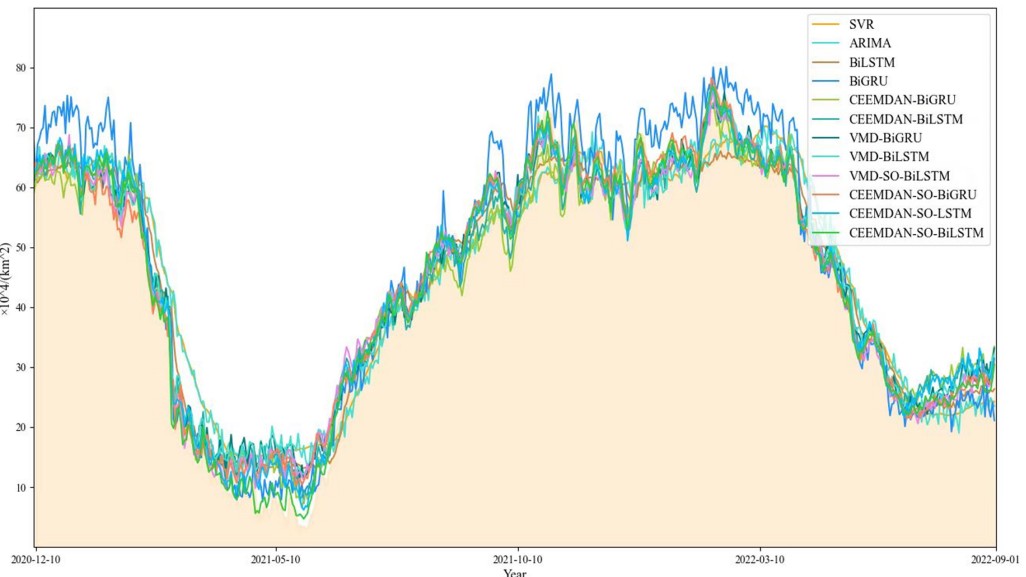

**Figure 8  Comparison of model prediction results.**

ARIMA model demonstrates a remarkable shortcoming in fitting the original time series during these periods.

On the other hand, the proposed CEEMDAN-SO-BiLSTM model yields a superior fitting capability, effectively capturing the variability of the time series. As evidenced in Tables 2 and 3, the proposed method exhibits remarkable advantages over both the single-model counterparts and the benchmark SVR model. Specifically, the combined CEEMDAN-SO-BiLSTM model reduces MAE, RMSE, and SMAPE by approximately 81.8%, 81.5%, and 9.150%, respectively, and raises R$^2$ by approximately 3.9%.

Furthermore, our experimental findings reveal that applying decomposition and optimization strategies consistently enhances prediction accuracy, with the CEEMDAN-SO-BiLSTM model outperforming its VMD counterpart. Finally, among the different

variants of the LSTM and BiLSTM models, CEEMDAN-SO-BiLSTM provides the best performance. These results underscore the efficacy and robustness of our proposed framework in accurately predicting sea ice areas in the Greenland Sea.

## CONCLUSION

In this article, a combined CEEMDAN-SO-BiLSTM model is proposed for predicting the daily sea ice area in the Greenland Sea. By decomposing the data into multiple relatively stable eigenmodes *via* the CEEMDAN method, the model takes into account the nonstationarity and nonlinear characteristics of the time series. By optimizing the hyperparameters of the BiLSTM model using the SO algorithm and training each mode separately, the final predictions are then merged and reconstituted to yield daily sea ice area forecasts.

An array of comparative experiments was conducted against alternative hybrid models to evaluate the effectiveness and practicability of the proposed approach. Experimental results indicate that CEEMDAN decomposition considerably enhances the extraction of relevant features from the time series and leads to reduced RMSE and MAE predictions by 2.201 and 0.297, respectively, compared to the single BiLSTM model. Moreover, the hyperparameter optimization through SO strengthens the sensitivity of the CEEMDAN-BiLSTM model to data perturbations, resulting in improved evaluation metrics, including MAE, RMSE, and SMAPE, with respective reductions of 1.768, 2.201, and 7.910% and an increase of 2.20% in $R^2$.

Despite these promising findings, there remain some limitations due to insufficient data leading to suboptimal hyperparameter settings and the lack of interpretability in deep learning models hindering exhaustive error analysis. Future research directions may include integrating additional environmental factors, exploiting advanced deep learning structures such as GNNs or attention mechanisms, enhancing data quality and quantity through methods such as data fusion and augmentation, and addressing issues related to interpretability and error diagnosis. Ultimately, advancements along these lines will enable the development of increasingly accurate and applicable sea ice prediction models.

### Funding
The authors received no funding for this work.

### Competing Interests
The authors declare there are no competing interests.

### Author Contributions
- Qiao Guo conceived and designed the experiments, performed the experiments, analyzed the data, prepared figures and/or tables, authored or reviewed drafts of the article, and approved the final draft.

- Haoyu Zhang conceived and designed the experiments, performed the experiments, analyzed the data, prepared figures and/or tables, authored or reviewed drafts of the article, and approved the final draft.
- Yuhao Zhang conceived and designed the experiments, performed the experiments, analyzed the data, prepared figures and/or tables, authored or reviewed drafts of the article, and approved the final draft.
- Xuchu Jiang conceived and designed the experiments, performed the experiments, analyzed the data, prepared figures and/or tables, authored or reviewed drafts of the article, and approved the final draft.

## Data Availability

The data and code are available in the Supplemental Files.

The dataset used in this article is available from the sea ice data set on the official website of the National Environmental Information Center (https://www.ncei.noaa.gov/).

## Supplemental Information

Supplemental information for this article can be found online at http://dx.doi.org/10.7717/peerj.15748#supplemental-information.

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
