# Peer review of "Prediction of sea ice area based on the CEEMDAN-SO-BiLSTM model"

_PeerJ, doi:10.7717/peerj.15748_

## Round 0.1 · original submission · Major Revisions

The 2 reviews are quite different and both sets of comments needs to be addressed and reconciled.

Reviewer 1 brings up the issue of the training and testing periods that needs to be addressed. They also raise the issue of timescales, stating that on longer timescales, simpler methods using inertia and so on might be appropriate. The paper illustrates short-term timescales but also mentions trends, so its focus is unclear.

Any revision needs to target specific timescales, rather than making general statements about forecasting. By being more focused, the paper can address the 7 major points raised by the reviewer. The authors finish the paper by stating that more work is needed on prediction and interpretation, but surely it is important to show here where the authors think these benefits lie, by giving an initial example. Any new method may require investment of time and resources compared to existing methods (even in learning how to apply it), so potential costs and benefits need to be flagged.

Reviewer 2 also points to this lack of focus, but is more positive about the techniques applied. Accordingly, any revision will need to apply a tighter focus from the introduction to the conclusion. It is well understood that the prediction of sea ice is important, so saying where and how this paper is addressing that issue needs to be made clearer.

Finally, the manuscript needs a lot of work to be publishable. Both reviewers mention issues with notation, equations and references. The paper also needs a lot of work on its language. Engaging a professional or very experienced writer of English is recommended. More active and concise language is recommended.

Reviewer 1 ·

Basic reporting

English expression is not professional, it is recommended to find professionals for improvement.

Experimental design

The experimental setup is reasonable but simple, and it is recommended to add relevant experiments to prove the validity of the prediction model from different perspectives.

Validity of the findings

no comment

Additional comments

See specific comments!

Annotated reviews are not available for download in order to protect the identity of reviewers who chose to remain anonymous.

Reviewer 2 ·

Basic reporting

Comment 1: In Section 2.1, line 11, the variable i is written in both regular and italic font, which may cause confusion. It is recommended to use only one consistent font style throughout the formula, preferably italic.

Comment 2: In Section 2.1, line 9, the variable n appears in the equation without explanation, which may be confusing for readers. It is suggested to provide explanations for all variables used in the equations.

Comment 3: In Section 3.2, the formulas for evaluation metrics RMSE, MAE, R2, and SMAPE are not consistently formatted, and it is recommended to modify them to use regular font and include an explanation of the variable n.

Comment 4: In Section 3.1, line 15, specify the data website more clearly.

Comment 5: Please change BILSTM to BiLSTM.

Experimental design

Comment 1: The abstract lacks emphasis on the main contributions and does not provide a summary of the conclusions. It is recommended to clarify the main findings of the study. The Conclusion lacks quantitative comparisons and explicit statements about model performance, making the conclusions less clear. It is recommended to revise this section accordingly.

Comment 2: Figure 3 has inconsistent colors compared to Figures 1 and 2, and it is suggested to improve the overall visual coherence of the figures. The font of the legends in Figures 6, 7, and 8 is not consistent. It is recommended to change it to Times New Roman font.

Comment 3: References [8], [10], [11], and [15] contain outdated methods and conclusions that are not suitable for the literature review section. It is suggested to update them with more recent studies.The second and fourth paragraphs of the Introduction rely too heavily on references and lack clear analysis and organization. It is recommended to revise them for better coherence and clarity of logic.

Validity of the findings

The main contribution of this paper is the combination of frequency domain decomposition algorithm, swarm intelligence optimization, and BiLSTM neural network for predicting sea ice area. The empirical study has shown that the proposed model outperforms the baseline models such as BiLSTM and VMD-BiLSTM, demonstrating the superiority of adaptive noise complete set empirical mode decomposition. I find the topic and structure of this manuscript interesting and timely. If the following modifications can be made, I recommend publishing it.

---

## Round 0.2 · Minor Revisions

Dear authors,
You need to describe the data used. It is missing from the paper. You may also want to add some additional context relating your model to the real world as per my comments in the previous review.

Reviewer 2 ·

Basic reporting

no comment

Experimental design

no comment

Validity of the findings

no comment

Additional comments

Thank you for the author's response. After inspection, I found that the author has made modifications one by one according to my suggestions and provided point-to-point responses. The new version has been greatly improved, and I am satisfied with it. There are currently no other new opinions.

---

## Round 0.3 · accepted · Accept

Thank you for addressing the review comments. The manuscript is now ready for publication.